

# Facile imine synthesis under green conditions using Amberlyst® 15

Giovanna Bosica, Roderick Abdilla, Kaylie Demanuele and Josef Fiteni

Department of Chemistry, University of Malta, Msida, Malta

## ABSTRACT

Imines and their derivatives are of great interest to organic synthetic chemists due to their involvement as key intermediates which facilitate the construction of nitrogen heterocycles, particularly the formation of alkaloids. Imine formation by condensation of primary amines with aromatic aldehydes and cyclohexanone has been investigated under environmentally-friendly solventless heterogeneous catalysis. An array of different imines was obtained in excellent yields in appreciably short reaction times using Amberlyst® 15 as a heterogeneous catalyst. The latter was used owing to its high commercial availability, recyclability, ease of separation from the reaction mixture, and versatility.

## INTRODUCTION

Imines are fundamental intermediates in the synthesis of N-containing organic molecules which are biologically active (particularly alkaloids) or used in various industrial procedures (*Kataja & Masson, 2014*). Their formation *via* the condensation of amines with aldehydes or ketones is a reversible process (*Kataja & Masson, 2014*). In fact, the use of imines as intermediates in multicomponent reactions is very ubiquitous due to the same equilibrium which exists (*Jin et al., 2014*). Imine-based multicomponent reactions which have gained considerable attention in the past few years include: formation of quinolizidines and indolizidines *via* A$^3$ coupling, other Mannich-type reactions (nitro-Mannich, aza-Friedel-Crafts, Petasis), and the intramolecular aza-Diels-Alder reaction (*Rajesh, Purohit & Rawat, 2015*; *Kumar et al., 2014*; *Ramesh & Nagarajan, 2011*).

In order to be able to drive the condensation process to completion, the classical method required azeotropic distillation by a Dean-Stark apparatus which obviously necessitated the use of excess amounts of solvents (*Darling, 1966*). Subsequently, alternative eco-friendly procedures were developed including the Schmidt reaction, the oxidative-dehydrogenation of amines and the oxidative coupling of alcohols and amines just to name a few (*Adimurthy & Patil, 2013*). Such procedures still may suffer from one or more of the following disadvantages: require expensive metallic catalysts, have a lower atom economy, require long reaction times, require the use of toxic solvents and show overall low environmental benignity (*Darling, 1966*; *Adimurthy & Patil, 2013*).

After the incorporation of the green chemistry principles into the synthetic chemistry modus operandi, following the works of *Anastas & Warner (1998)*, the above mentioned

Corresponding author
Giovanna Bosica,
giovanna.bosica@um.edu.mt

syntheses were improved further and rendered more eco-friendly through the use of heterogeneous catalysts such as: sulfated-$TiO_2$, montmorillonite K-10 clay, sulfated nano-ordered silica and zeolites (*Kumar et al., 2016*; *Atanassova et al., 2011*; *Naimi-Jamal, Ali & Dekamin, 2013*; *Amrutham et al., 2017*). Further recently developed sustainable methodologies have been reported using irradiation techniques (*Rizzuti et al., 2020*). Moreover the addition of dehydrating agents such as phosphorous-pentoxide-silica have also been shown to drive the reaction to completion by removing the condensation product, ergo water (*Naeimi et al., 2008*). Obviously such agents require oven drying and reactivation for future re-use.

Another difficulty associated with this synthetic reaction is that in general, ketones are much less reactive when compared with aldehydes rendering ketimine synthesis even more unfavourable and non-green.

In the following research, in continuation of our interest in the application of heterogeneous catalysts (*Bosica, 2021*), herein we report how various heterogeneous catalysts and some desiccants were tried and compared in terms of activity and efficiency for the imine synthesis reaction. Amberlyst® 15, a cheap commercially available catalyst, is found to give the highest yields in short reaction times under neat conditions. In addendum, the unprecedented synthesis of the ketimine from the condensation of cyclohexanamine and cyclohexanone proved successful.

## EXPERIMENTAL

### Materials used

All commercially available chemicals were purchased from Aldrich and used without further purification.

### Instrumentation

For the characterization of final products and monitoring of the reactions the same procedure described in detail in our previous publications was followed in order to obtain FTIR, NMR and MS and GC spectra (*Bosica & Abdilla, 2017*).

### General procedure

The general procedure for the imine-synthesis reaction involved stirring the aldehyde (5 mmol) and the amine (5.5 mmol) in the presence of 0.2 g of A15 catalyst under neat conditions at room temperature in a nitrogen-dried 25 mL one-neck round bottom flask. The reaction was monitored *via* both TLCs and/or GC analysis. The catalyst was filtered off by suction and washed appropriately with diethyl ether solvent (approximately 5–10 mL). The filtrate was concentrated by rotary evaporation and by a double-stage vacuum oil pump in order to remove the unreacted amine for reactions involving low-boiling amines. The products of aromatic amines were purified by recrystallization from ethanol or by column chromatography using a 9:1, 8:2 or 7:3 hexane/ethyl acetate eluant ratio. The TLC plates used for monitoring were composed of silica on PET with fluorescent indicator. Plates were observed under a UV lamp at a wavelength of 254 nm

**Scheme 1 Model reaction between benzaldehyde (1a) and *t*-butylamine (2a).**

(5h, 85%)

**Scheme 2 Condensation of cyclohexanamine (2g) with cyclohexanone (1i) in the presence of Amberlyst® 15 catalyst at room temperature using an amine excess of 0.1 equivalent.**

before staining in an iodine-saturated chamber. Analytical data for all products are reported in the Supporting Information File.

## RESULTS AND DISCUSSION

During the initial screening, the reaction between benzaldehyde (Scheme 1a) and *t*-butylamine (Scheme 2a) was chosen as the model reaction (Scheme 1). The latter was always performed at room temperature under quasi-solvent-free conditions in the presence/absence of a number of heterogeneous catalysts and desiccants. In addition, the molar ratios of the reagents and catalyst quantities were also varied. Table 1 shows the results of the preliminary catalyst/drying agent screening trials.

As evidenced, out of the catalysts and desiccants tried and tested, the best yields were obtained using Montmorillonite K-10, Amberlyst® 15 and acidic alumina. However, the work up of the reactions involving either MK-10 or acidic alumina required the addition of more solvent than that involving Amberlyst® 15 owing to their physical state (powder). Henceforth, Amberlyst® 15 (being in the form of macroporous beads) was selected for the subsequent optimization trials especially considering its ease of separation from the reaction mixture. In addition, it was discovered that a smaller amount of Amberlyst® 15 could result in even higher yields (Table 2) possibly owing to easier mechanical stirring and less product adsorption onto the catalyst beads.

One of the main limitations of the condensation of primary amines and aldehydes/ketones is the equilibrium which exists between the products and the substrates. This explains why in the initial trials the amine was used in excess, ergo, to shift the equilibrium

**Table 1 Optimization trials involving various heterogeneous catalysts.**

| Entry[a] | Catalyst (Amount in g) | Reaction time (h) | Yield[b] 3a (%) |
|---|---|---|---|
| 1[c] | N/A | 9.5 | 81 |
| 2 | Anhydrous $MgSO_4$ (1.00) | 8 | 88 |
| 3 | Molecular sieves (3Å)[d] (0.50) | 6.5 | 98 |
| 4 | Molecular sieves (4Å)[d] (0.50) | 4 | 94 |
| 5 | Amberlyst® A-21 (1.50) | 2 | 78 |
| 6 | Amberlyst® 15 (1.50) | 2 | 86 |
| 7 | Montmorillonite K-10 (0.30)[e] | 2.5 | 95 |
| 8 | Nafion® SAC-13 (0.40)[f] | 2 | 78 |
| 9 | Acidic alumina (1.25) | 2 | 97 |
| 10 | Neutral alumina (1.25) | 2.5 | 75 |
| 11 | Basic alumina (1.25) | 2 | 49 |
| 12 | 25% w/w KF on basic alumina (1.00) | 2.0 | 79 |
| 13[h] | 1.3 mmol% CuI on Amberlyst® A-21 (0.05) | 1.5 | 80[g] |

**Notes:**
[a] All reactions were carried out in the presence of 1 mL of diethyl ether on a 5 mmol scale using a 1:3 molar ratio of 1a:2a at room temperature (*circa* 15–25 °C). The addition of diethyl ether was required because reaction mixture soon thickened significantly after reaction initiation.
[b] Yield of pure isolated product unless otherwise indicated.
[c] No desiccant or catalyst added.
[d] Effective pore size in Angstrom.
[e] A smaller amount of MK-10 (compared to other catalysts) was used because on addition of larger amounts of MK-10, reaction mixture dried up immediately and the addition of 1 mL of diethyl ether was not enough to aid stirring.
[f] When a larger amount of catalyst was used (1.5 g), the Nafion beads kept moving out of reaction mixture and adhering to reactant flask walls.
[g] Reaction carried out under solventless conditions owing to the small amount of catalyst.
[h] Copper iodide leaching was noted due to residual green colour in the crude reaction mixture following catalyst filtration.

**Table 2 Optimization trials involving changing the amount of Amberlyst® 15.**

| Entry[a] | Quantity of Amberlyst® 15 (g) | Reaction time (h) | Yield[b] 3a (%) |
|---|---|---|---|
| 1 | 0.6 | 2 | 98 |
| 2 | 0.4 | 2 | 99 |
| 3 | 0.2 | 2 | 99 |
| 4 | 0.1 | 2.5 | 99 |
| 5 | 0.05 | 3 | 97 |

**Notes:**
[a] All reactions were carried out under solventless conditions at room temperature on a 5 mmol scale using a benzaldehyde:*tert*-butylamine molar ratio of 1:3.
[b] Yield of pure isolated product.

**Table 3 Optimization trials involving varying the reagent ratios.**

| Entry[a] | Aldehyde:amine molar ratio | Reaction time (h) | Yield[b] 3a (%) |
|---|---|---|---|
| 1 | 1:2 | 2 | 99 |
| 2 | 1:1.5 | 2 | 99 |
| 3 | 1:1.1 | 2 | 99 |

**Notes:**
[a] All reactions carried out under solventless conditions at room temperature on a 5 mmol scale.
[b] Yield of pure isolated product.

**Table 4 Condensation of primary aliphatic amines with aromatic aldehydes using Amberlyst® 15 as catalyst.**

| Entry[a] | Aldehyde/Ketone | Amine | Product | Time (h) | Yield (%)[b] |
|---|---|---|---|---|---|
| 1 | 1a | 2a | 3a | 2 | 99 |
| 2 | 1b | 2a | 3b | 3 | 96 |
| 3 | 1c | 2a | 3c | 4.5 | 93 |
| 4 | 1d | 2a | 3d | 4.5 | 98 |
| 5 | 1e | 2a | 3e | 3 | 97 |
| 6 | 1f | 2a | 3f | 2 | 97 |

(Continued)

| Entry[a] | Aldehyde/Ketone | Amine | Product | Time (h) | Yield (%)[b] |
|---|---|---|---|---|---|
| **Table 4** (continued) | | | | | |
| 7[c] | 1g | 2a | 3g | 80 | NY[d] |
| 8 | 1h | 2a | 3h | 80 | NY[d] |
| 9 | 1a | 2b | 3i | 1 | 99 |
| 10 | 1i | 2b | 3j | 1 | 97 |
| 11 | 1c | 2b | 3k | 3 | 93 |
| 12 | 1d | 2b | 3l | 2.5 | 92 |
| 13[c] | 1a | 2c | 3m | 1 | 98 |
| 14[c] | 1a | 2d | 3n | 1 | 99 |

**Notes:**
[a] All reactions carried out under solventless conditions at RT using 0.2 g Amberlyst® 15 on a 5 mmol scale using an amine equivalent excess of 0.1.
[b] Yield of pure isolated product collected after work-up and purification.
[c] Reactions carried out in a 1:1.5 aldehyde/ketone:amine molar ratio.
[d] No pure product collected.

forward. Yet, as outlined in Table 3, the yields remained practically the same even when the latter mentioned excess was decreased to 0.1 equivalents only.

Subsequently, having identified the ideal conditions (neat, room temperature, 0.2 g per 5 mmol Amberlyst® 15, 0.1 equivalent excess of amine), the substrate scope could be expanded by varying the aldehydes and the amines. In general, the best outcomes were obtained (Table 4) when using aromatic aldehydes and aliphatic primary amines due to the

**Table 5** Condensation of aromatic primary amines with aromatic aldehydes.

| Entry[a] | Aldehyde | Amine | Product | Time (h) | Yield (%)[b] |
|---|---|---|---|---|---|
| 1 | 1a | 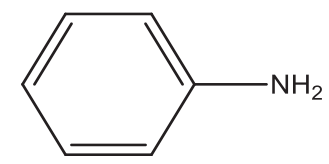 2e | 3o | 3 | 72 |
| 2 | 1b | 2e | 3p | 2.5 | 81 |
| 3 | 1c | 2e | 3q | 4 | 75 |
| 4[c] | 1e | 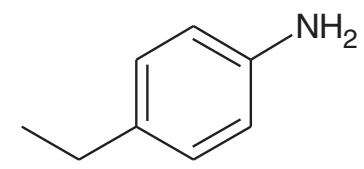 2f | 3r | 3 | 90 |

**Notes:**
[a] All reactions carried out under solventless conditions using 0.2 g Amberlyst® 15 on a 5 mmol scale using an aldehyde/ketone:amine molar ratio of 1:1.1.
[b] Yield of pure isolated product collected by recrystallization from ethanol.
[c] Reactions carried out using a 1:1.5 aldehyde/ketone:amine ratio.

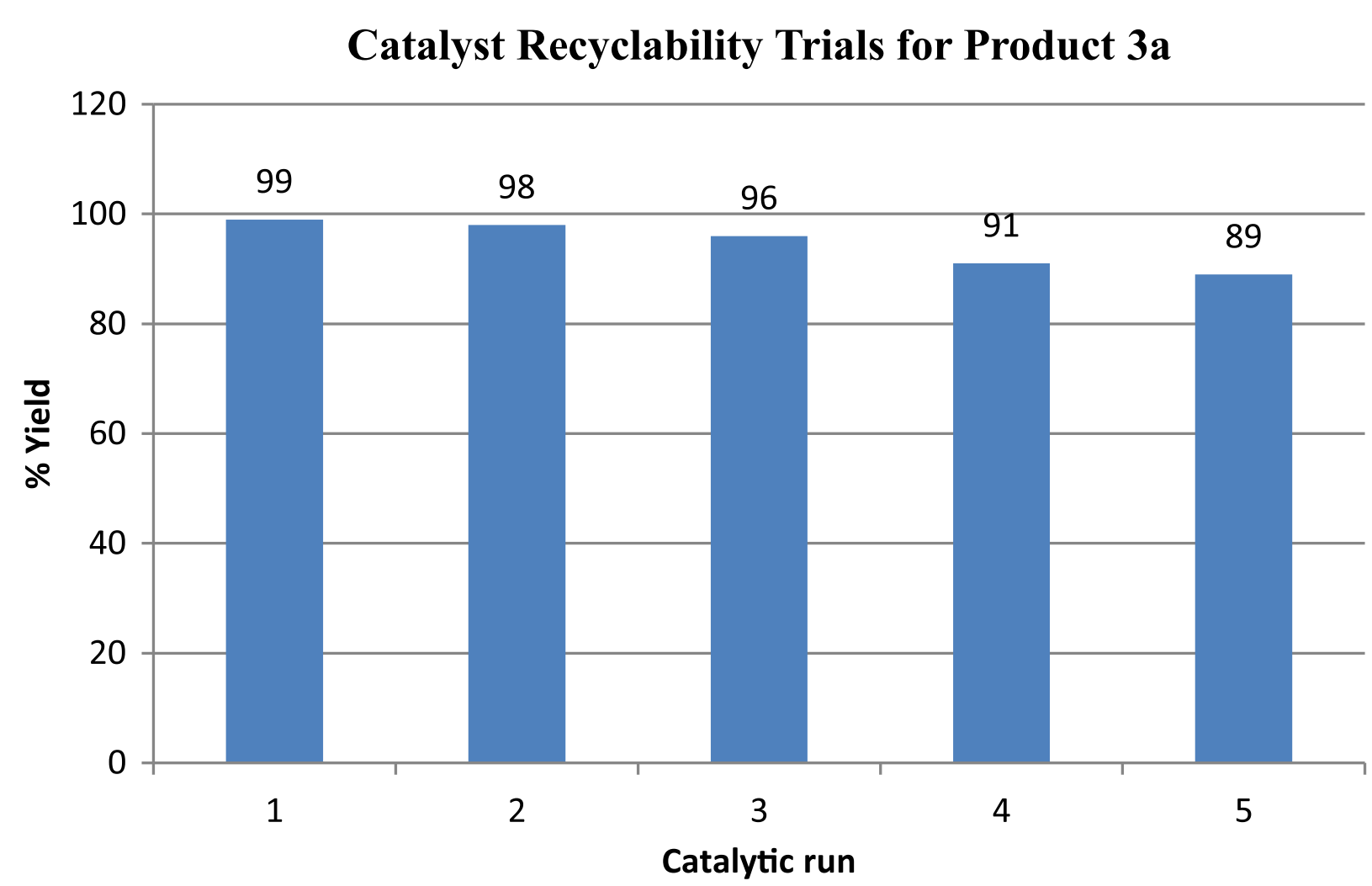

**Figure 1** Amberlyst® 15 catalyst recycling trials for model reaction (to form product 3a).

higher reactivity of the aldehydes as opposed to ketones and the greater nucleophilic character of aliphatic amines as opposed to aromatic amines.

Positively, despite their inherent lack of reactivity, aromatic amines also gave appreciable yields as outlined in Table 5. Not only, but the primary aliphatic amine, c-hexylamine, was able to react successfully with the cyclic ketone, cyclohexanone (Scheme 2) to give the product (Scheme 3s) in 85% yield.

Lastly, the catalyst exhibited good recyclability because the model reaction could be repeated up to five times with the same catalyst with the yield decreasing by 10% between the first and last trial (Fig. 1). The latter decrease is most probably a result of sulfonic acid group inactivation by the reaction with the amine reactant.

## CONCLUSIONS

The heterogeneous, safe-to-handle, relatively-cheap and commercially-available Amberlyst® 15 was found to be the ideal catalyst for the synthesis of various imines using both aliphatic and aromatic amines and aromatic aldehydes (72–99% yields, 17 examples) in significantly short reaction times (2–4 h) at room temperature in neat conditions. The catalyst morphology, *i.e.*, being in the form of macroporous beads, enabled it to be easily recovered with minimal solvent use during work up and reused for up to five times.

## ACKNOWLEDGEMENTS

The authors would like to thank Prof. Robert M. Borg for assistance with the acquisition of the NMR spectra and Dr. Godwin Sammut for MS analyses.

### Funding

The authors received no funding for this work.

### Competing Interests

Giovanna Bosica is an Academic Editor for PeerJ.

### Author Contributions

- Giovanna Bosica conceived and designed the experiments, analyzed the data, prepared figures and/or tables, authored or reviewed drafts of the article, and approved the final draft.
- Roderick Abdilla analyzed the data, prepared figures and/or tables, authored or reviewed drafts of the article, and approved the final draft.
- Kaylie Demanuele performed the experiments, authored or reviewed drafts of the article, and approved the final draft.
- Josef Fiteni performed the experiments, prepared figures and/or tables, and approved the final draft.

### Data Availability

Raw data is available as a Supplemental File.

### Supplemental Information

Supplemental information for this article can be found online at http://dx.doi.org/10.7717/peerj-ochem.7#supplemental-information.

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

## FURTHER READING

**Bisht R, Chattopadhyay B. 2016.** Formal Ir-catalyzed ligand-enabled ortho and meta borylation of aromatic aldehydes via in-situ generated imines. *Journal of the American Chemical Society* **138(1)**:84–87 DOI 10.1021/jacs.5b11683.

**Blackburn L, Taylor RJK. 2001.** In-situ oxidation-imine formation-reduction routes from alcohols to amines. *Organic Letters* **3(11)**:1637–1639 DOI 10.1021/ol015819b.

**Donthiri RR, Patil RD, Adimurthy S. 2012.** NaOH-catalyzed imine synthesis: aerobic oxidative coupling of alcohols and amines. *European Journal of Organic Chemistry* **2012(24)**:4457–4460 DOI 10.1002/ejoc.201200716.

**Ibert Q, Cauwel M, Glachet T, Tite T, Martel PL, Lohier J, Renard P, Franck X, Reboul V, Sabot C. 2021.** One-pot synthesis of diazirines and $^{15}N_2$-diazirines from ketones, aldehydes and derivatives: development and mechanistic insight. *Advanced Synthesis & Catalysis* **363(18)**:4390–4398 DOI 10.1002/adsc.202100679.

**Ivanov DP, Dubkov KA, Babushkin DE, Pirutko LV, Semikolenov SV. 2010.** Liquid-phase hydroamination of cyclohexanone. *Russian Chemical Bulletin* **59(10)**:1896–1901 DOI 10.1007/s11172-010-0330-x.

**Okimoto M, Takahashi Y, Numata K, Nagata Y, Sasaki G. 2005.** Electrochemical oxidation of benzylic amines into the corresponding imines in the presence of catalytic amounts of KI. *Synthetic Communications* **35**:1989–1995 DOI 10.1081/SCC-200066648.

**Reddy MM, Kumar MA, Swamy P, Naresh M, Srujana K, Satyanaryana L, Venugopal A, Narender N. 2013.** N-alkylation of amines with alcohols over nanosized zeolite beta. *Green Chemistry* **15(13)**:3474–3483 DOI 10.1039/c3gc41345d.

**Sain B, Jain SL, Singhal S. 2007.** [Bmim]$BF_4$-immobilized Rhenium-catalyzed highly efficient oxygenation of aldimines to oxaziridines using solid peroxides as oxidants. *Journal of Organometallic Chemistry* **692**:2930–2935 DOI 10.1016/j.jorganchem.2007.03.004.