# Peer review of "Facile imine synthesis under green conditions using Amberlyst® 15"

_PeerJ Organic Chemistry, doi:10.7717/peerj-ochem.7_

## Round 0.1 · original submission · Major Revisions

Dear Prof. Bosica

Your manuscript was evaluated by three experts in organic synthesis. As you can see, one of the reviewers did not recommend publication. Still, I'm willing to offer you the opportunity to make the adjustments recommended by the reviewers. As soon as you are in a position to submit a new version, I will be happy to re-evaluate your work.

Reviewer 1 ·

Basic reporting

The work is written confidentially

Experimental design

Could be interesting to use also aliphatic aldehydes or other aliphatic ketones in addition to the cyclic one 1i.

Validity of the findings

The procedure has several advantages including simplicity, irreversibility and the green and environmentally friendly conditions (room temperature, recyclable catalyst). Moreover immines are important intermediates in the synthesis of biologically active N-heterocyclic products and in industrial synthetic processes.

Additional comments

The work reports a simple and efficient procedure to obtain imines. A variety of aromatic aldehydes containing electron-withdrawing and electron-donating groups, with aliphatic and aromatic ammines can be employed; could be interesting to use also aliphatic aldehydes or other aliphatic ketones in addition to the cyclic one 1i.
I think that the manuscript can be published in PeerJ because it has several advantages including irreversibility and the green and environmentally friendly conditions (room temperature, recyclable catalyst).
Some typing errors:
In the line 15 of abstract is reported “aliphatic and aromatic aldehydes” change with “aromatic aldehydes” and also in line 120 change “aldehydes” with “aromatic aldehydes”.
In line 43 the reference could be added: “Green Chemistry Theory and Practice Paul T. Anastas and John C. Warner Oxford University Press 1998”
Line 108: change “Scheme 1” with “Scheme 2”
Line 185: change “Subash, B.” with “Subash, B.;”
Line 195: change “Kodumuri, S.” with “Kodumuri, S.;” and “Gajula, K.S.” with “Gajula, K.S.;”
Schemes and tables:
note a of Table 2: change “benzaldehyde:amine” with “1a:2a”
Scheme 2: change “5hrs” in “5h”
Supporting information:
line 25: change “748 ” with “748. ”
lines 52 and 72: change “H NMR” with “1H NMR”
lines 80: superscript 1 in “1H NMR”
put lines 82 and 83 (white solid and IR) before line 81

Reviewer 2 ·

Basic reporting

Dr. Giovanna Bosica and coauthors described the synthesis of imine by condensation of primary amines with aliphatic and aromatic aldehydes and cyclohexanone under environmentally-friendly solventless heterogeneous catalysis. They reported that Amberlyst® 15 is a performant heterogeneous catalyst in this transformation. Furthermore, the commercial availability and recyclability of this catalyst, the ease of separation from reaction mixture and versatility, makes this synthetic process an innovative and interesting approach for the synthesis of imine derivatives.
The paper is generally well referenced but, in my opinion, when critical points are reported to describe the disadvantages of other procedures (rows 38-41), related references should be added.
Anyway, some changes to improve the quality of this paper are required.
Preliminary experiments (table 1 and 2) should be more detailed, and the related results reported in just one table (Table 1 + Table 2).
The header of all tables (all named as "table 1") and the caption of the schemas (all named as "Scheme 1") need to be corrected.
Cyclic aliphatic amines have a nitrogen comprised in the ring (i.e., piperidine, pyridine, etc....), cyclohexylamine is just a primary aliphatic amine with a cyclic substituent (row 103).
To evaluate the effectiveness of using this amine, the reaction with cyclohexanone must also be compared with other amines. Further data could be obtained using cyclohexylamine also with other carbonyl substrates.
Instead of “catalyst’s physical state” (row 121), the term “catalyst morphology” is preferable.
In reference 2, an author is missed (Wei Zhang). Please check.
In reference 11, the article title is “Hβ Catalyzed condensation reaction between aromatic ketones and anilines: To access ketimines (imines)”. Please check.
Some changes in the Supporting Information file are also required.
Some signals that refer to a singlet are reported as multiplet and/or with ppm expressed as a range (i.e., methoxy and tBu groups, etc….), why? (For example, see compounds 3c, 3d, 3e, 3f, 3k ….).
The 1H-NMR data of compound 3i are reported in reverse order of all the others.
The 1H-NMR data reported for compound 3j, in my opinion, don’t fit structural characteristics; furthermore, the indicated reference (18), refer to the 4-nitro derivative NMR data; please check.
Compound 3k was described with one more aromatic hydrogen (5 ArH instead of 4 ArH); a comma is missed before 8.20 (row 49).
Compound 3m: a comma instead of “and” is required (row 59).
Compound 3p: the J for peaks at 8.21 and 8.70 ppm are missed (row 73).
Compound 3r: (row 81), please erase the bracket after Hz; IR data should be moved before NMR data; the signal of iminic hydrogen is probably missed.

Experimental design

No comment

Validity of the findings

The authors have to check carefully NMR data and the way they have to be reported. Scanned spectra could be added to the Supporting information file in order to help the reader.

Reviewer 3 ·

Basic reporting

The manuscript titled “Facile imine synthesis under green conditions using Amberlyst® 15” describes the synthesis of sixteen aldimines obtained from de reaction of arylaldehydes with alkyl/aryl amines catalyzed by Amberlyst® 15 at room temperature under solventless conditions. Still, one example of a ketimine is presented. In my opinion, the reaction concept is well known as previously published by several authors (J. Org. Chem., 1971, 86, 1570, Synthesis 1985; 679, Tetrahedron Letters 1997, 38, 2039, Synlett 2004, 2135, J. Chem. Res. 2005, 299, Synthesis 2006, 1652, etc). I have several observations that justify why the manuscript cannot be accepted for publication on PeerJ Journal at this moment. Please, see the following points.
1) I did not fully understand from the Table 1 and 2 why Amberlyst-15 was chosen as the optimal catalyst. Overall yields of 3a in the case of acid alumina, M K-10, molecular sieves, and anhydrous MgSO4 were higher.
2) For me, the section “Results and Discussion” is poor, and it is limited to aryl aldehydes and few amines.
3) General Procedure: The purification method needs to be better elucidated. Which products were purified by column chromatography? Which were crystallized? Details on crystallization method need to be provided when appropriate.
4) One of the steps in typical procedures for the synthesis of compounds 3 is washing the reaction mixture with diethyl ether. Authors should add the information concerning the amount of diethyl ether used in this step.
5) Supporting information: The supporting information has twenty-one references with no need.
6) Authors should thoroughly check the values of spin-spin coupling constants through all the Supporting Information taking into account the fact that the spin-spin coupling constants of the protons, which are coupled to each other, should be the same.
7) The melting point analysis is missing for compounds 3o, 3p and 3r.

Experimental design

No comment.

Validity of the findings

Totally, I don’t think this work has big impact for organic synthesis research. The synthesized compounds are known and the manuscript lacks novelty. So, I do not recommend this manuscript to be published in PeerJ.

---

## Round 0.2 · Minor Revisions

Thanks for providing the revised version of your manuscript. Before a final decision, however, I would like to request special attention to what was raised by Reviewer 2, regarding the characterization of compound 3j and the spectra figures in SI material.

Reviewer 1 ·

Basic reporting

The revised manuscript considered most of the reviewers’ comments. The Tables have been corrected and more literature data has been added. Despite the unfortunate failure of the instrument, which did not allow the insertion of the preliminary experiments with other aldehydes/aliphatic ketones, many substrates are reported and characterized.

Experimental design

I think that the manuscript is suitable for publication in Peer J in the present form because represents a very simple, eco-friendly, efficient, and alternative approach to the synthesis of imines.

Validity of the findings

The ease of use and recoverability of Amberlyst 15® makes the catalyst a viable cheap alternative

Reviewer 2 ·

Basic reporting

Dr. Giovanna Bosica and coauthors submitted the revised manuscript “Facile imine synthesis under green conditions using Amberlyst® 15” accordingly to reviewer’s suggestions.
Unfortunately, there are still problems with the 1H-NMR data reported for compound 3j.
The chemical shifts of aliphatic protons do not correspond to the isobutyl group of 3j and they are in absolute disagreement with those reported in reference 20.
In my opinion, the derivative 3j is neither the N-(3-nitrobenzylidene)-2-methylpropan-1-amine nor N-(4-nitrobenzylidene)-2-methylpropan-1-amine. The authors have to check the starting reagents and repeat the experiment.
Furthermore, I did not find copies of selected scanned spectra in the Supporting information file.
Row 73: round bottom flask.
Row 155: (ref 2) please add a semicolon between P. and Zhang.

Experimental design

no comment

Validity of the findings

no comment

Additional comments

no comment

Reviewer 3 ·

Basic reporting

In my previous review, several were the comments made to the authors. The authors have properly revised the manuscript and SI which are better now.
Therefore, I would recommend accepting this manuscript in its current form.

Experimental design

No comment.

Validity of the findings

No comment.

Additional comments

No comment.

---

## Round 0.3 · accepted · Accept

Dear Dr. Bosica,
Thank you for submit the revised version of your manuscript.
I am pleased to inform you that the revised version of the above-mentioned paper has been accepted for publication in PeerJ Organic Chemistry. Your paper will now be processed for production.